# Efficacy and Safety of TiNO-Coated Stents versus Drug-Eluting Stents in Acute Coronary Syndrome: Systematic Literature Review and Meta-Analysis

**DOI:** 10.3390/biomedicines10123159

**Published:** 2022-12-07

**Authors:** Frederic C. Daoud, Louis Létinier, Nicholas Moore, Pierre Coste, Pasi P. Karjalainen

**Affiliations:** 1INSERM, BPH, U1219, Bordeaux University, 33000 Bordeaux, France; 2Coronary Care Unit, Cardiologic Hospital, Bordeaux University, 33604 Pessac, France; 3Cardiac Unit, Heart and Lung Center, Helsinki University Hospital, Helsinki University, 00280 Helsinki, Finland

**Keywords:** meta-analysis, acute coronary syndrome, percutaneous coronary intervention, drug-eluting stents, non-drug-eluting titanium-nitride-oxide coated stents (TiNOS), Major Adverse Cardiac Events (MACE), recurrent myocardial infarction, stent thrombosis

## Abstract

(1) Background: Practice guidelines define drug-eluting stents (DES) as the standard of care in coronary percutaneous coronary intervention (PCI), including in acute coronary syndrome (ACS). This is based on comparisons with bare-metal stents (BMS). However, non-drug-eluting titanium-nitride-oxide-coated stents (TiNOS) have not been taken into account. The objective of this study is to determine whether TiNOS can be used as an alternative to DES in ACS. (2) Methods: A prospective systematic literature review (SLR), conducted according to the PRISMA guidelines, was performed, wherein multiple literature databases from 2018 and 2022 were searched. Prospective, randomised, controlled trials comparing outcomes after PCI with TiNOS vs. DES in any coronary artery disease (CAD) were searched. Clinical outcomes were meta-analytic pooled risk ratios (RR) of device-oriented Major Adverse Cardiac Events (MACE) and their components. The analysis stratified outcomes reported with ACS-only vs. ACS jointly with chronic coronary syndrome (CCS). (3) Results: Five RCTs were eligible, comprising 1855 patients with TiNOS vs. 1363 with DES at a 1-year follow-up. Three enrolled patients presented with ACS only and two with ACS or CCS. The latter accounted for most of the patients. The one-year pooled RRs in those three RCTs were as follows: MACE 0.93 [0.72, 1.20], recurrent myocardial infarction (MI) 0.48 [0.31, 0.73], cardiac death (CD) 0.66 [0.33, 1.31], clinically driven target lesion revascularization (TLR) 1.55 [1.10, 2.19], and stent thrombosis (ST) 0.35 [0.20, 0.64]. Those results were robust to a sensitivity analysis. The evidence certainty was high in MACE and moderate or low in the other endpoints. (4) Conclusions: TiNOS are a non-inferior and safe alternative to DES in patients with ACS.

## 1. Introduction

Coronary Artery Disease (CAD) is classified into two broad groups: Acute Coronary Syndrome (ACS) and chronic coronary syndromes (CCS) [1]. CCS have a variety of presentations but are without acute symptoms.

Patients with ACS have ongoing acute myocardial ischemia that can cause various symptoms ranging from cardiac arrest to electrical or haemodynamic instability and cardiac mechanical disorders. The leading ACS symptom is chest discomfort or pain. ACS with acute chest pain and persistent (>20 min) ST-segment elevation on an electrocardiogram (ECG) often reflects an acute total or subtotal coronary occlusion. Most patients with this type of ACS develop ST-segment elevation myocardial infarction (STEMI) [2]. Patients with acute chest discomfort and no persistent ST-segment elevation (NSTE-ACS) often develop non-ST-segment elevation myocardial infarction (NSTEMI), but some of them do not develop myocardial damage and are classified as having unstable angina (UA). The published evidence has established percutaneous coronary interventions (PCI) with drug-eluting stents (DES) as the standard of care in CAD, including the different presentations of ACS [2,3,4,5,6,7,8,9,10]. The purpose of using DES is to mitigate post-stenting restenosis [11,12,13]. Titanium-nitride-oxide-coated coronary stents (TiNOS) have a pharmacologically inactive, non-absorbable coating. Preclinical studies have shown less neointimal hyperplasia with TiNOS than with bare-metal stents (BMS) [14,15,16]. Several randomised, controlled clinical trials (RCTs) have compared clinical outcomes of TiNOS vs. DES in patients with ACS. Pooling the results to summarise them and enable an overall interpretation is now required. This study is the first systematic literature review (SLR) on this topic. Its objective is to determine if TiNOS can be used as an alternative to DES in ACS.

## 2. Materials and Methods

### 2.1. Foundations

This SLR was designed and conducted according to methods described in the Cochrane Handbook with the use of “Grading of Recommendations Assessment, Development and Evaluation” (GRADE) [17,18,19,20,21,22]. The protocol was registered in PROSPERO (CRD4201809062) before initiation. It is reported according to “Preferred Reporting Items for Systematic Reviews and Meta-Analyses” (PRISMA) [23].

### 2.2. Research Question Specification

The research question was specified using the PICOS framework and Academic Research Consortium (ARC-2) definitions [24,25]. Patients presented ACS. The intervention was PCI using TiNOS. The comparator was PCI using any DES. Outcomes constituted the device-oriented Major Adverse Cardiac Events (MACE), a composite of three items: Cardiac death (CD), recurrent myocardial infarction (MI), and clinically driven target lesion revascularisation (TLR). CD or MI was summarised jointly to reflect the safety component of MACE with a single indicator. In addition, Probable or definite ST was analysed as a stent-related serious adverse event (SAE) that may result in CD or MI. All-cause mortality (TD) was also analysed. All measures of outcome were assessed at one- and five-year follow-ups. The study methods were parallel-arm prospective RCTs.

### 2.3. Data Sources

Pubmed, Embase, the Cochrane Library, and Web of Science (WoS) electronic databases were queried on 8 March 2018 and 27 August 2022. The search terms were: *((bioactive OR (Titanium AND nitride AND oxide) OR TiNO OR TNO OR BAS) AND stent) AND (DES OR (drug AND eluting AND stent)) AND (RCT OR ((randomised OR randomised) AND controlled AND trial))*. No exclusion filter was applied related to language, country, year, or any other aspect. The search string as interpreted by the databases’ search engines is reported in Appendix B. The websites of AHA, TCT, ESC, EuroPCR, and clinicaltrials.gov were also searched for unpublished studies meeting the question’s specifications.

The downloaded record files were imported, pooled, and sifted in EndNote X8 (Clarivate Analytics, Philadelphia, PA, USA). One reference only was selected when duplicates were identified. When different references concerned the same study, their information was pooled using the citation of the most recent one. Full articles were reviewed for all non-duplicate references.

### 2.4. Study Selection, Risk of Bias Analysis, Data Extraction, Certainty of Evidence Grading

Two reviewers (FD and LL) independently performed the following steps: (1) exhaustive reference screening, (2) reference classification according to the inclusion and exclusion criteria, (3) extraction of study methods, (4) of patient baseline data, (5) of treatment data, and results of each eligible RCT, (6) individual RCT risk of bias rating, and (7) assessment of the certainty of the evidence for each outcome variable according to GRADE [19,20,21,22]. The screened studies were included if they met the following criteria: provided first-hand clinical evidence with prospective inclusion; included patients with CAD treated with coronary PCI; involved implantation of either TiNOS or DES after the random allocation of the stent type; provided target outcomes reported at 1-year and/or 5-year follow-up; provided the outcomes reported as the number of patients who were included along with the number or proportion of them who presented an event of interest.

RCTs were included if IRB/ethics committee’s approval and patients’ informed consent were confirmed, the evidence was first-hand with prospective inclusion, patients were treated for CAD treated with coronary PCI, stents were either TiNOS or DES after the random allocation, target outcomes were reported at 1-year and/or 5-year follow-up as the number of patients included, and events were reported with the number or proportion of included patients. ACS refers to patients presenting ST-elevated myocardial infarction (STEMI), non-ST-elevated myocardial infarction (NSTEMI), or unstable angina pectoris at baseline as defined in ARC-2 [25]. Data extraction was stratified according to patients’ clinical presentation, i.e., ACS vs. CCS, whenever feasible. Differences were adjudicated by a third reviewer (NM). One reviewer (PK) adjudicated multiple endpoint definitions across the RCTs. The results were recorded in Review Manager software (RevMan version 5.4.1, The Nordic Cochrane Centre, Copenhagen, Denmark), and the risk of bias of individual RCTs was rated according to the criteria proposed by the Cochrane Collaboration and was implemented via that software package with operator’s blinding as an additional separate item [26].

### 2.5. Meta-Analysis

The two treatment arms were compared concerning each endpoint using the risk ratio (RR) defined as ((n patients with an event in TiNOS)/(n patients in TiNOS))/((n patients with an event in DES)/(n patients in DES)). RR > 1 reflected a higher frequency of events in the TiNOS arm than in the DES arm and vice versa. Outcomes were analysed on an intention-to-treat (ITT) basis, with the number of patients presenting an event counted in the randomised arm as the numerator and the sample size of the corresponding arm as the denominator. Each RR was reported with its 95% confidence interval (CI []).

Publication bias was suspected if the funnel plot of study RRs was asymmetrical and/or if Harbord’s regression test for binary variables was significant (i.e., *p* < 0.05) [27,28,29,30]. The set of pooled study RRs was considered homogeneous if Cochran’s Q-test was not significant (given the χ² distribution’s degrees of freedom df = k − 1 where k is the number of study RRs) and if the I² was low to moderate (i.e., I² = (Q − df)/Q × 100% ≤ 25%) [27,28,30,31,32,33,34]. Due to clinical heterogeneity concerning clinical indications and stent generations, pooled RRs were calculated using the M-H method with a random-effects model.

Pooled analyses were stratified according to patient enrollment, i.e., RCTs with ACS-only vs. RCTs with ACS and CCS without separate outcome reporting. Each pooled measure of outcomes was presented in a Forest plot. One-year and five-year outcomes were analysed separately.

Sensitivity analysis was performed by iteratively recalculating the pooled RR after removing one eligible RCT.

The certainty of the pooled evidence was rated in GRADEpro GDT software 2022 online (https://gradepro.org) [35]. Additional analyses were performed in STATA (version 16, StataCorp LP, College Station, TX, USA) using the metan and metaprop packages.

## 3. Results

### 3.1. Study Identification and Selection

The studies’ identification, screening, and selection are described in the PRISMA flowchart (Figure 1). One hundred and eighteen references were identified and nine publications with first-hand data about five RCTs were eligible for inclusion in the meta-analysis.

### 3.2. Eligible Study Characteristics

Five RCTs were eligible, as their baseline characteristics complied with the PICOS specification (Table 1).

The studies reported 1-year follow-up data for 1855 patients in the TiNOS arm vs. 1363 in the DES arm. The quantities of available patients at 5-year follow-up were 783 vs. 773.

Three RCTs reported outcomes in patients with ACS only (TITAX-AMI, BASE-ACS, and TIDES-ACS). The two others reported outcomes in patients presenting ACS or CCS without stratification: TIDE enrolled 143 patients with ACS (47%) and TITANIC-XV enrolled 112 (64.7%).

### 3.3. Publication Bias

The funnel plot and *the* Harbord test (*p* = 0.263) did not detect a risk of publication bias regarding the RR of MACE in all CAD cases at 1-year follow-up (Figure 2).

Similar conclusions were found for the other six endpoints at the 1-year follow-up in all CAD cases.

### 3.4. Individual Study Bias

The compiled risk of bias across the studies (Figure 3) shows an overall risk of bias that is generally less than 75%, except for operator blinding. The individual RCT risk of bias is reported with the pooled 1-year MACE RR (Figure 4).

There were some differences in the definitions of MACE and MI between studies, but they were applied similarly in both treatment arms.

### 3.5. Pooled Outcome Risk Ratios

The stratified, pooled RRs of the primary endpoint, 1-year MACE (i.e., ACS-only vs. ACS and CCS, and total), show no significant risk difference between TiNOS and DES and low individual study risk bias except for operator blinding (Figure 4). The ACES only vs. ACS and CCS subgroups display no significant heterogeneity according to the Q-test (*p* = 0.17), which is in line with the overlapping CIs. The 47.1% I² between subgroups quantifies the visual difference.

The pooled 5-year MACE RR shows a significantly lower pooled RR favouring TiNOS in the ACS-only subgroup. Heterogeneity between the two strata is larger than at one 1-year (I² = 66%), but the confidence intervals overlap with a non-significant Q-test (*p* = 0.09) (Figure 5).

The pooled RRs of recurrent non-fatal MI or CD at 1-year and 5-year follow-ups are significant and favour TiNOS in the ACS-only subgroup (Figure 6 and Figure 7). The RRs are driven by the differences in the incidence of non-fatal MI.

The pooled RR of TLR at a 1-year follow-up is significant, with more frequent TLRs with TiNOS than DES in ACS-only vs. ACS and CCS (Figure 8). However, the RR at 5-year follow-up reduces to non-significance as the rate of TLRs in the DES arm catches up with the rate in the TiNOS arm (Figure 9).

The pooled RRs of CD, non-fatal MI, probable or definite ST, TD, and definite ST are reported in the sensitivity analysis (Table 2). The 1-year and 5-year RRs of non-fatal MI and probable or definite ST are significant and favour TiNOS in the ACS-only subgroup.

The Forest plots of additional endpoints are in Appendix A: Recurrent non-fatal MI—1 year; Appendix A: Recurrent non-fatal MI—5 years; Appendix A: CD—1 year; Appendix A: CD—5 years; Appendix A: Probable or definite ST—1 year; Appendix A: Probable or definite ST—5 years.

### 3.6. Sensitivity Analysis—Additional Endpoints

The sensitivity analysis (Table 2) in the ACS-only group at the 1-year follow-up shows:-The robustness of the non-significant MACE’s pooled RR;-The robustness of the significant recurrent non-fatal MI’s pooled RR with less frequent events in TiNOS than in DES;-The robustness of the significant probable or definite ST’s pooled RR with less frequent events in TiNOS than in DES;-The robustness of the significant TLR’s pooled RR with more frequent events in TiNOS than in DES;-The robustness of CD’s pooled RR varied depending on the excluded study.

The total pooled RRs at the 1-year follow-up yielded similar results to ACS-only.

The sensitivity analysis in the ACS-only group at the 5-year follow-up also yielded similar results to ACS at the 1-year follow-up; however, the TLR RR became non-significant.

Sensitivity analysis in the total pooled RRs at the 5-year follow-up yielded similar results to ACS at the 5-year follow-up except for probable or definite ST, which lost robustness upon TITAX-AMI’s removal. The RR of definite ST remained robust.

### 3.7. GRADE: Certainty of the Evidence

Given the potential bias caused by pooling ACS and CCS, and considering that the 5-year outcomes of TIDES-ACS were not published when writing this article, the GRADE analysis focused on the 1-year outcomes in the ACS-only group (Table 3).

The certainty level of the evidence of MACE, the primary outcome measure, is high. However, the certainty levels of the secondary outcome measures are moderate, low, or very low.

The detailed explanations supporting each GRADE item rating are reported in Appendix C.

## 4. Discussion

Practice guidelines establish the use of DES as the standard of care in PCI to treat ACS [2,9,10]. However, the 2017 Cochrane review comparing the outcomes with DES vs. BMS in ACS concluded that the *“evidence in this review was of low to very low quality, and the true result may depart substantially from the results presented in this review*” [45]. Therefore, the comparison of the clinical outcomes of TiNOS vs. DES in ACS is relevant to determining whether TiNOS could be an alternative to DES in that group of clinical indications.

The first hypothesis of this SLR is based on preclinical demonstrations that TiNOS are associated with less neointimal hyperplasia than BMS [14,15,16]. The second hypothesis is that patients presenting with ACS have a higher risk of SAE than patients with CCS. Given that some RCTs included both patients with ACS and CCS, the SLR analysed all RCTs that compared DES to TiNOS.

The pooled RRs of MACE, CD or recurrent non-fatal MI, clinically-driven TLR, ST, and TD at the 1-year follow-up, did not significantly differ between all-RCTs and ACS-only RCTs. The latter represented 85% of the patients, and the ACS-only pooled results drove the all-RCT results. Therefore, the internal and external validation of this meta-analysis focuses on the ACS-only subgroup.

All the results were robust to the sensitivity analysis, so no single trial significantly modified them. This confirmed that excluding any ACS-only trial due to the generation of the platform, the eluted drug (e.g., paclitaxel), or a specific patient risk factor (e.g., diabetes) had no impact on the pooled results.

The results at the 5-year follow-up were consistent with those at the 1-year follow-up, although probable ST presented reduced robustness while definite ST remained robust.

No risk of publication bias was identified, and the overall risk of bias in the individual RCTs was not serious except for non-blinding operators.

At the 1-year follow-up in the ACS-only trials, TiNOS and DES displayed a non-significantly different MACE rate, and the quality of evidence was high. TiNOS displayed a significantly lower rate of CD or recurrent MI than DES, and the quality of evidence was moderate due to lower precision caused by the limited number of events observed altogether. TiNOS and DES displayed a non-significantly different TLR. TiNOS displayed lower mortality and ST rates, and the quality of evidence was low or very low with fewer observed events.

Two previously published meta-analyses comparing DES vs. BMS in ACS were identified to attempt the external validation of the DES arm of this meta-analysis [45,46]. The 2017 Cochrane review included 25 RCTs, of which most focused on STEMI, with different time horizons and RRs that were reported at the maximum follow-up. Therefore, a comparison with the 1-year and 5-year outcomes of this meta-analysis was not interpretable. The 2022 individual patient data meta-analysis (IPDM) includes 14 RCTs with a total of 22,319 patients with 34.5% of the patients treated for CCS and 65.5% for ACS. The outcomes are reported at 1-year and 5-year follow-ups. The type of ACS reported in the source publications (Table 4) shows 36.2% STEMI, 33.6% NSTEMI, and 33.6% unstable angina compared with the 42% STEMI, 47% NSTEMI, and 34% unstable angina reported in the publications of the three ACS-only RCTs of this meta-analysis. The IPDM pooled data concerned 14,628 ACS cases with 7739 DES vs. 6889 BMS (details in the IPDM Appendix A). The IPDM and this meta-analysis show relatively similar pooled rates of CD or MI in DES at the 1-year follow-up (ratio: 0.89) and 5-year follow-up (ratio: 1.35) (Table 5). This comparability supports the validity of that endpoint in this meta-analysis. However, the pooled incidence rate of the definite ST in the DES group of this meta-analysis is 3 times higher at the 1-year follow-up than in the IPDM and 5.48 times higher at the 5-year follow-up. These ratios could have resulted from differences in the methods, the patients’ baseline risk, the types of DES used, and the relatively small number of observations in this meta-analysis. Moreover, the IPDM does not report TLR or MACE rates, so IPDM does not provide an external validation basis for those three endpoints.

Overall, the robustness to the sensitivity analysis, the low level of heterogeneity between the three ACS-only RCTs, and their low risk of bias support the internal validity of this meta-analysis of TiNOS vs. DES. The similarity in the CD or MI rates between the DES arms of the IPDM and this meta-analysis support the external validity of that endpoint. One can infer from the meta-analyses that DES presents a lower risk of CD or MI than BMS and TiNOS has a lower risk than DES.

This meta-analysis shows with high certainty of evidence, according to GRADE, that TiNOS is non-inferior to DES in ACS at the 1-year follow-up. TiNOS displays a significantly lower risk of recurrent non-fatal myocardial infarctions and probable or definite ST but a significantly higher risk of TLR. The certainty of evidence concerning ST and TLR is low due to the limited number of observations, thus the limited precision in the GRADE criteria. At any rate, the risk of TLR with TiNOS can be reduced when using short stent lengths (≤28 mm) and/or large stent diameters (>3.0 mm) [42,44].

The results at the 5-year follow-up are consistent with the 1-year results but will require the publication of TIDES-ACS final data to be confirmed.

## 5. Conclusions

This systematic review shows that titanium-nitride-oxide-coated stents are non-inferior to drug-eluting stents when applied to acute coronary syndrome at the one-year follow-up in terms of device-oriented major adverse cardiac events and present a lower risk of recurrent non-fatal myocardial infarction. The interim five-year results are consistent with the one-year results and are robust. Therefore, titanium-nitride-oxide-coated stents are a safe alternative to drug-eluting stents in acute coronary syndrome.

Note: Ineligible records are listed with the references [61,62,63,64,65,66,67,68,69,70,71,72,73,74,75,76,77,78,79,80,81,82,83,84,85,86,87,88,89,90,91,92,93,94,95,96,97,98,99,100,101,102,103,104,105,106,107,108,109,110,111,112,113,114,115,116,117,118].

## Figures and Tables

**Figure 1 biomedicines-10-03159-f001:**
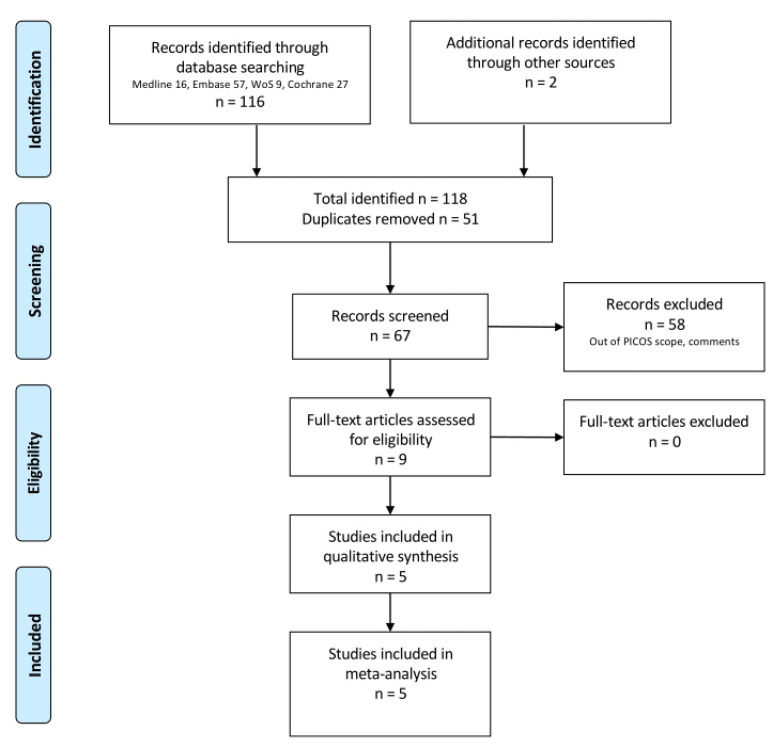
PRISMA flow chart.

**Figure 2 biomedicines-10-03159-f002:**
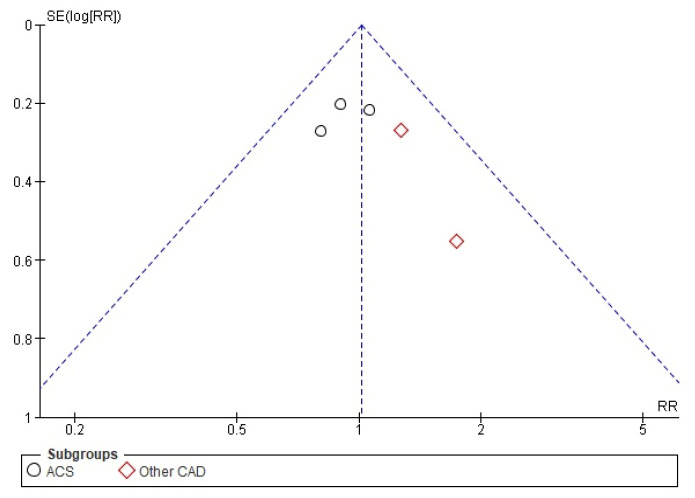
Publication bias of MACE RR in all RCTs at 1-year follow-up: funnel plot.

**Figure 3 biomedicines-10-03159-f003:**
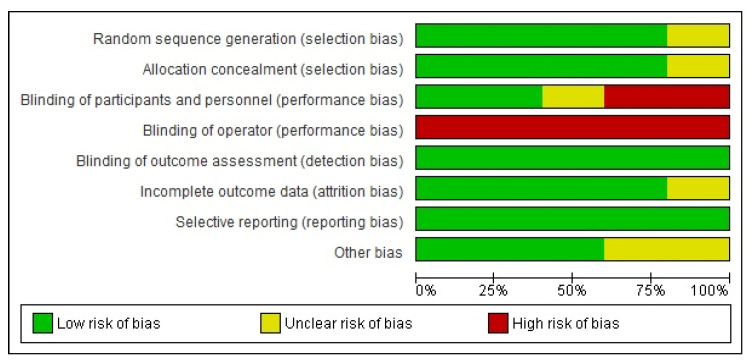
Individual RCT risk of bias across studies.

**Figure 4 biomedicines-10-03159-f004:**
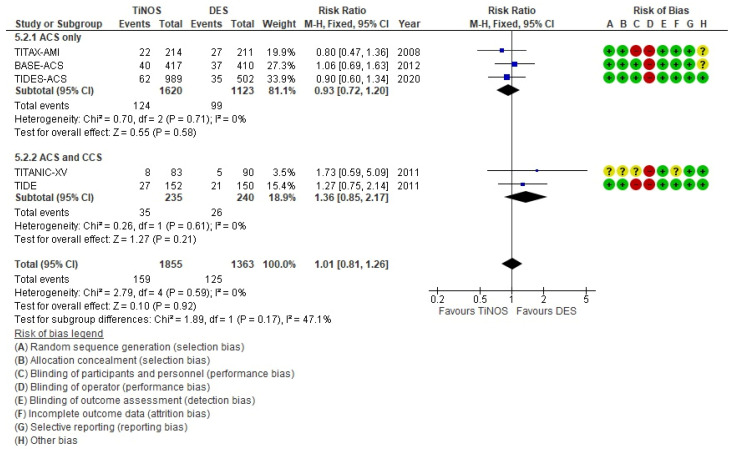
MACE—1 year.

**Figure 5 biomedicines-10-03159-f005:**
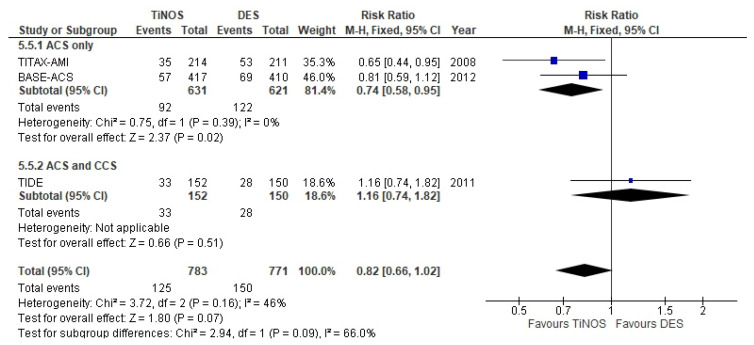
MACE—5 years.

**Figure 6 biomedicines-10-03159-f006:**
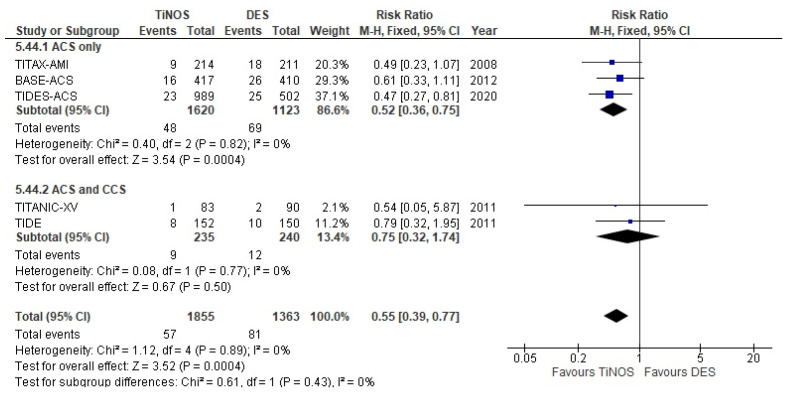
CD or MI—1 year.

**Figure 7 biomedicines-10-03159-f007:**
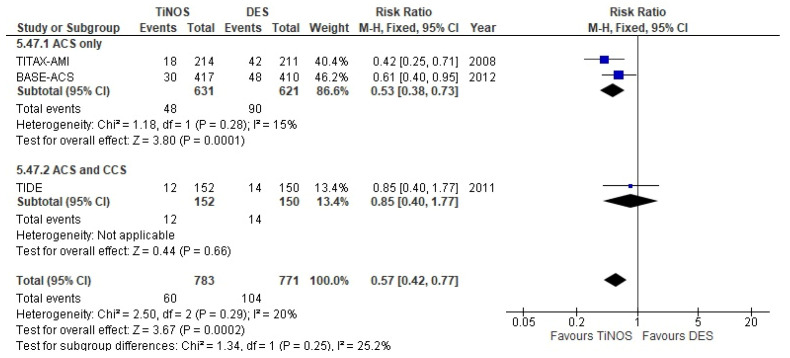
CD or MI—5 years.

**Figure 8 biomedicines-10-03159-f008:**
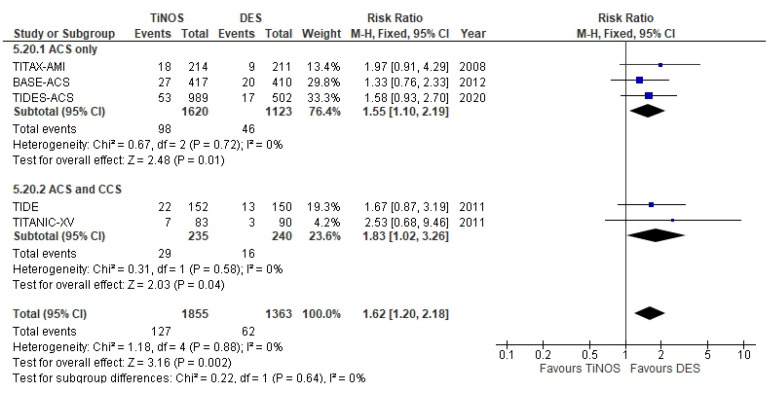
TLR—1 year.

**Figure 9 biomedicines-10-03159-f009:**
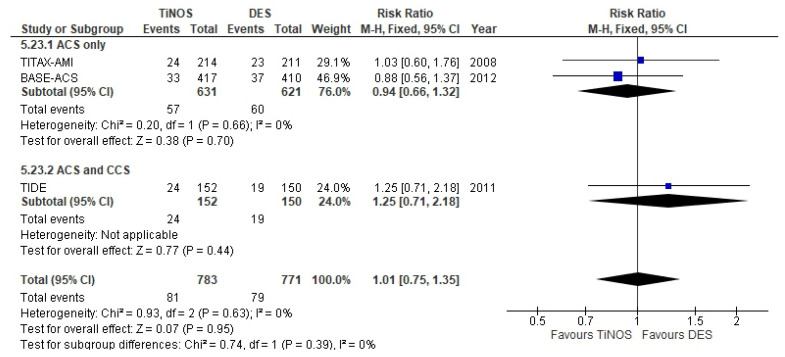
TLR—5 years.

**Table 1 biomedicines-10-03159-t001:** Eligible studies’ baseline characteristics.

Study	Age and Prior Events			Clinical Presentation (Included and Pooled)			Procedural Data and Medication			Attrition, Cross-Overs
Stent		TiNOS	DES		TiNOS N incl, % pooled	DES N incl, % pooled		TiNOS	DES	
TITAX-AMI DES = PES [36,37]	Patients n age prior MI prior PCI prior CABG	214 64 ± 11 15% 10% 7%	211 64 ± 11 9% 5% 6%	NSTEMI STEMI UA	131, 61% 83, 39% 0, 0%	114, 54% 97, 46% 0, 0%	Stents/culprit lesion n TSL (mm) post-dilation procedural success DAPT 12 m	1.1 ± 0.3 18.5 ± 6.4 42% 99.5% 31%	1.1 ± 0.4 19.2 ± 7.2 35% 98.1% 65%	LFU 1 y TiNOS: 0 DES: 0 5 y TiNOS 3 DES: 7
TIDE DES = ZES [38,39]	Patients n age prior MI prior PCI prior CABG	152 65.9 ± 9.0 27.6% 25.7% 7.9%	150 63.4 ± 10.5 28.7% 21.3% 25.3% 2.7%	Stable angina NSTEMI UA	57.9% 32.9% 9.2%	47.3% 42.0% 10.7%	Stents/culprit lesion n TSL (mm) device success DAPT 12 m	1.28 ± 0.55 19.3 ± 11.1 93.0%	1.17 ± 0.45 19.6 ± 10.0 94.6%	LFU 1 y TiNOS: 0 DES: 2 5 y: N.R.
TITANIC-XV DES = EES [40]	Patients n age prior MI prior PCI prior CABG	83 66.5 ± 8.8 10.8% 8.4% 2.4%	90 64.5 ± 10.1 15.6% 11.1% 2.2%	NSTEMI other CAD Diabetics: all	69.9% 30.1%	60.0% 40.0%	Stents/culprit lesion n TSL (mm) stent failure DAPT 12 m	1.1 ± 0.3 18.72 ± 8.20 NR.	1.1 ± 0.3 21.63 ± 9.65 NR.	NR.
BASE-ACS DES = EES [41,42]	Patients n age prior MI prior PCI prior CABG	417 63 ± 12 13.4% 9.6% 4.8%	410 63 ± 12 9.8% 10.5% 4.1%	NSTEMI STEMI UA	206, 49.4% 162, 38.8% 49, 11.8%	187, 45.6% 159, 38.8% 64, 15.6%	Stents/culprit lesion n TSL (mm) post-dilation stent failure DAPT: Aspirin: N.R. Clopidogrel: N.R.	1.15 ± 0.38 20.8 ± 9.4 42.2% 0.0%	1.14 ± 0.36 20.6 ± 8.2 43.9% 1.0%	LFU 1 y TiNOS: 3 DES: 3 5 y TiNOS 29 DES: 28
TIDES-ACS DES = EES [43,44]	Patients n age prior MI prior PCI prior CABG	989 62.7 ± 10.7.6% 7.0% 0.6%	502 62.6 ± 10.5 9.0% 6.6% 1.2%	NSTEMI STEMI	46.3% 44.9%	45.0% 47.6%	Stents/culprit lesion n TSL (mm) post-dilation stent failure DAPT 12 m	1.13 ± 0.38 20.5 ± 7.8 33.0% 0.3% 80.3%	1.14 ± 0.37 20.6 ± 7.2 38.0% 1.0% 86.0%	LFU 1 y TiNOS: 7 DES: 4

UA = unstable angina pectoris; Drugs: EES = everolimus elution; ZES = zotarolimus elution; PES = paclitaxel elution; 12 m = 12 months; 1 y = 1 year.

**Table 2 biomedicines-10-03159-t002:** Sensitivity analysis of primary and secondary endpoints.

	M-H Fixed Effects RR and 95% CI after the Removal of:					
Endpoint	None	TITAX-AMI	TIDE	TITANIC-XV	BASE-ACS	TIDES-ACS
Total (all RCTs)—1-year follow-up						
MACE	1.01 [0.81, 1.26]	1.06 [0.83, 1.36]	0.96 [0.75, 1.23]	0.98 [0.78, 1.24]	0.99 [0.76, 1.29]	1.07 [0.82, 1.40]
CD or MI	0.55 [0.39, 0.77]	0.57 [0.39, 0.82]	0.52 [0.36, 0.74]	0.55 [0.39, 0.77]	0.53 [0.36, 0.79]	0.60 [0.40, 0.91]
Non-fatal MI	0.52 [0.36, 0.76]	0.51 [0.33, 0.78]	0.48 [0.31, 0.73]	0.52 [0.35, 0.76]	0.60 [0.38, 0.93]	0.51 [0.32, 0.82]
CD	0.66 [0.33, 1.31]	0.77 [0.37, 1.61]	0.66 [0.33, 1.31]	0.66 [0.33, 1.31]	0.30 [0.11, 0.81]	1.11 [0.43, 2.85]
Clinically driven TLR	1.62 [1.20, 2.18]	1.56 [1.13, 2.15]	1.60 [1.15, 2.24]	1.58 [1.16, 2.14]	1.74 [1.22, 2.47]	1.63 [1.14, 2.33]
Probable or definite ST	0.39 [0.22, 0.69]	0.46 [0.25, 0.84]	0.35 [0.20, 0.64]	0.39 [0.22, 0.69]	0.36 [0.18, 0.72]	0.38 [0.16, 0.87]
Definite ST	0.51 [0.26, 0.99]	0.49 [0.25, 0.97]	0.46 [0.23, 0.92]	0.51 [0.26, 0.99]	0.62 [0.29, 1.36]	0.52 [0.18, 1.44]
TD	0.78 [0.48, 1.27]	0.77 [0.46, 1.32]	0.78 [0.47, 1.27]	0.78 [0.48, 1.27]	0.49 [0.26, 0.95] ^b^	1.22 [0.65, 2.28]
ACS-only RCTs—1-year follow-up						
MACE	0.93 [0.72, 1.20]	0.97 [0.73, 1.30]	NA.	NA.	0.86 [0.63, 1.19]	0.95 [0.68, 1.33]
CD or MI	0.52 [0.36, 0.75]	0.53 [0.35, 0.80]	NA.	NA.	0.48 [0.30, 0.75]	0.56 [0.35, 0.90]
Non-fatal MI	0.48 [0.31, 0.73]	0.45 [0.27, 0.74]	NA.	NA.	0.55 [0.33, 0.92]	0.44 [0.25, 0.77]
CD	0.66 [0.33, 1.31]	0.77 [0.37, 1.61]	NA.	NA.	0.30 [0.11, 0.81]	1.11 [0.43, 2.85]
Clinically driven TLR	1.55 [1.10, 2.19]	1.46 [0.99, 2.15]	NA.	NA.	1.69 [1.09, 2.63]	1.53 [0.97, 2.40]
Probable or definite ST	0.35 [0.20, 0.64]	0.42 [0.22, 0.78]	NA.	NA.	0.32 [0.15, 0.65]	0.31 [0.12, 0.77]
Definite ST	0.46 [0.23, 0.92]	0.43 [0.21, 0.89]	NA.	NA.	0.54 [0.24, 1.24]	0.39 [0.12, 1.25]
TD	0.78 [0.47, 1.27]	0.77 [0.45, 1.32]	NA.	NA.	0.47 [0.24, 0.93] ^b^	1.23 [0.64, 2.35]
Total (all RCTs)—5-year follow-up—Interim Results						
MACE	0.82 [0.66, 1.02]	0.91 [0.70, 1.19]	0.74 [0.58, 0.95] ^b^	NA.	0.83 [0.62, 1.10]	Expected
CD or MI	0.57 [0.42, 0.77]	0.67 [0.46, 0.97]	0.53 [0.38, 0.73]	NA.	0.53 [0.35, 0.80]	Expected
Non-fatal MI	0.56 [0.39, 0.80]	0.59 [0.37, 0.92]	0.54 [0.37, 0.80]	NA.	0.57 [0.35, 0.93]	Expected
CD	0.68 [0.39, 1.19]	0.93 [0.47, 1.82]	0.59 [0.31, 1.11]	NA.	0.55 [0.25, 1.24]	Expected
Clinically driven TLR	1.01 [0.75, 1.35]	1.00 [0.71, 1.42]	0.94 [0.66, 1.32]	NA.	1.13 [0.77, 1.66]	Expected
Probable or definite ST	0.30 [0.14, 0.61]	0.45 [0.19, 1.05]	0.25 [0.12, 0.55]	NA.	0.22 [0.07, 0.70]	Expected
Definite ST	0.25 [0.11, 0.55]	0.37 [0.14, 0.99]	0.20 [0.09, 0.49]	NA.	0.22 [0.07, 0.70]	Expected
TD	1.03 [0.74, 1.45]	1.14 [0.75, 1.73]	0.95 [0.65, 1.37]	NA.	1.05 [0.65, 1.69]	Expected
ACS-only RCTs—5-year follow-up—Interim Results						
MACE ^a^	0.74 [0.58, 0.95]	0.81 [0.59, 1.12] ^c^	NA.	NA.	0.65 [0.44, 0.95] ^c^	Expected
CD or MI	0.53 [0.38, 0.73]	0.61 [0.40, 0.95]	NA.	NA.	0.42 [0.25, 0.71]	Expected
Non-fatal MI	0.54 [0.37, 0.80]	0.56 [0.33, 0.94] ^c^	NA.	NA.	0.53 [0.30, 0.94] ^c^	Expected
CD	0.59 [0.31, 1.11]	0.83 [0.38, 1.84] ^c^	NA.	NA.	0.33 [0.11, 1.00] ^c^	Expected
Clinically driven TLR	0.94 [0.66, 1.32]	0.88 [0.56, 1.37] ^c^	NA.	NA.	1.03 [0.60, 1.76] ^c^	Expected
Probable or definite ST ^a^	0.25 [0.12, 0.55]	0.37 [0.15, 0.93] ^c^	NA.	NA.	0.13 [0.03, 0.57] ^c^	Expected
Definite ST ^a^	0.20 [0.09, 0.49]	0.28 [0.09, 0.85] ^c^	NA.	NA.	0.13 [0.03, 0.57] ^c^	Expected
TD	0.95 [0.65, 1.37]	1.02 [0.63, 1.65] ^c^	NA.	NA.	0.85 [0.48, 1.53] ^c^	Expected

^a^ sensitivity analysis in ACS at 5-year follow-up results in the RR and confidence intervals of individual RCTs; ^b^ borderline shift with 3 or fewer RCTs contributing to the pooled estimate; ^c^ results based on a single-trial; NA—Not applicable.

**Table 3 biomedicines-10-03159-t003:** GRADE Summary of findings—TiNOS vs. DES in ACS at 1-year follow-up.

Outcome	Risk of Bias	Inconsistency	Indirectness	Imprecision	Publication Bias	Overall Certainty of Evidence
Device-oriented MACE	not serious ^a^	not serious ^b^	not serious ^c^	not serious ^d^	none	⨁⨁⨁⨁ HIGH
CD or MI	not serious ^a^	not serious ^b^	not serious ^e^	serious ^f^	none	⨁⨁⨁◯ MODERATE
Clinically driven TLR	not serious ^a^	not serious ^b^	not serious ^g^	very serious ^h^	none	⨁⨁◯◯ LOW
Non-fatal MI	not serious ^a^	not serious ^b^	not serious ^i^	serious ^j^	none	⨁⨁⨁◯ MODERATE
CD	not serious ^a^	very serious ^k^	not serious ^l^	very serious ^m^	none	⨁◯◯◯ VERY LOW
Probable or definite ST	not serious ^a^	not serious ^b^	not serious ^n^	very serious ^o^	none	⨁⨁◯◯ LOW
TD	not serious ^a^	serious ^k,p^	not serious ^q^	very serious ^r^	none	⨁◯◯◯ VERY LOW

**Explanations:** GRADE Working Group grades of evidence: High certainty—very confident that the true effect lies close to that of the estimate. Moderate certainty—moderately confident in the effect estimate; the true effect is likely to be close to the estimate of the effect, but there is a possibility that it is substantially different. Low certainty—confidence in the effect estimate is limited; the true effect may be substantially different from the estimate. Very low certainty—very little confidence in the effect estimate; the true effect is likely to be substantially different from the estimate.

**Table 4 biomedicines-10-03159-t004:** Outcome comparison of this meta-analysis with Piccolo et al. 2022 IPDM in ACS.

Study	N Total	N (%) with ACS
SPIRIT I [47]	56	Total: 9 (16.1%) unstable angina: 9
ENDEAVOR II [48]	1197	Total: 359 (30%) unstable angina: 359
PAINT [49]	491	Total: 90 (18.3%) unstable angina: 90
BASKET PROVE [50]	2314	Total: 1492 (64.5%) unstable angina: 754 STEMI: 738
CORACTO [51]	91	not reported
ISAR-CABG [52]	610	Total: 239 (39.2%) unstable angina: 239
PRODIGY [53]	2003	Total: 1465 (73.1%) unstable angina: 367 STEMI: 450 NSTEMI: 648
INSPIRON [54]	57	Total: 13 (22.8%) unstable angina: 13
XIMA [55]	740	not reported
BASKET PROVE II [56]	2291	Total: 1446 (63.1%) unstable angina: 787 STEMI: 659
LEADERS-FREE [57]	2442	Total: 1029 (42.1%) unstable angina: 370 NSTEMI: 554 STEMI: 105
ZEUS [58]	1606	Total: 1016 (63.3%) unstable angina: 270 NSTEMI: 441 STEMI: 305
NORSTENT [59]	9013	Total: 6319 (70.1%) unstable angina: 1105 NSTEMI: 2842 STEMI: 2372
SENIOR [60]	1200	Total: 544 (45.3%) unstable angina: 109 NSTEMI: 308 STEMI: 127

**Table 5 biomedicines-10-03159-t005:** Outcome comparison this meta-analysis with Piccolo et al. 2022 IPDM in ACS.

	Piccolo et al. 2022		Ratios between the Two Meta-Analyses	This SLR	
Outcome and follow-up	DES	BMS		DES	TiNOS
CD or MI 1-year	535	636		69	48
	7739	6889		1123	1620
	0.0691	0.0923		0.0614	0.0296
			*DES IPDM/DES here*	*0.89*	
CD or MI 5-year	831	892		90	48
	7739	6889		621	631
	0.1074	0.1295		0.1449	0.0761
			*DES IPDM/DES here*	*1.35*	
Definite ST 1-year	46	74		20	14
	7739	6889		1123	1620
	0.0059	0.0107		0.0178	0.0086
			*DES IPDM/DES here*	*3.00*	
			*DES IPDM/TiNOS here*		*1.45*
Definite ST 5-year	66	91		29	6
	7739	6889		621	631
	0.0085	0.0132		0.0467	0.0095
	0.0085	0.0132	*DES IPDM/TiNOS here*	*5.48*	
			*DES IPDM/TiNOS here*		*1.11*

## Data Availability

Not applicable.

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
