# Peer review of "Efficacy and Safety of TiNO-Coated Stents versus Drug-Eluting Stents in Acute Coronary Syndrome: Systematic Literature Review and Meta-Analysis"

_biomedicines, 2022, doi:10.3390/biomedicines10123159_

Round 1

Reviewer 1 Report

This paper is very interesting, however, there is some problems.

Authors have concluded that TiNOS could be used as a first-line stent in patients with acute coronary syndrome and a high risk of complications associated with drug-eluting stents. Please describe in detail what are the high risk of complications associated with DESs and what the benefits of the TiNOS stent are.

Author Response

This paper is very interesting, however, there is some problems.

Authors have concluded that TiNOS could be used as a first-line stent in patients with acute coronary syndrome and a high risk of complications associated with drug-eluting stents. Please describe in detail what are the high risk of complications associated with DESs and what the benefits of the TiNOS stent are.

Authors’ reply: It was a unjustied interpretation of findings based on the current positioning of BMS used only in patients who are allergic to the eluted drug or present a high-risk of bleeding with dual antiplatelet treatment. The concept of “high risk” does not apply here and been removed altogether.

Reviewer 2 Report

The topic of this review is interesting and the authors claim this is the first study of its class, hence increasing its relevance. The methodology of the meta-analysis is detailed and described well in the following steps. The introduction is ok, but in my point of view, a better way to present data is needed with the exception of fig1 to fig 3. The rest of the results are difficult to follow

Comments

1. For the abstract section, an objective of the work is needed to add after the background section

2.- There are too many forest plots. It is possible to make a new figure resume the most important information of these plots?

3.- More references needs to be used in the discussion section and more explication of the results needs to be added

4.- The conclusion section does not express any particular conclusion or any new information obtained after the discussion of this work, I suggest rewriting the conclusion and making it more useful for understanding your contribution to this area of knowledge

5.- I suggest adding more references, from the last 5 to 10 years in the discussed topics, upgrading the number to more than 60 references

Author Response

The topic of this review is interesting and the authors claim this is the first study of its class, hence increasing its relevance. The methodology of the meta-analysis is detailed and described well in the following steps. The introduction is ok, but in my point of view, a better way to present data is needed with the exception of fig1 to fig 3. The rest of the results are difficult to follow

Comments

  1. For the abstract section, an objective of the work is needed to add after the background section

Authors’ reply: Done. The objective is stated at the end of the background in the abstract and the body of the article.

2.- There are too many forest plots. It is possible to make a new figure resume the most important information of these plots?

Authors’ reply: We managed to move half the Forest plots to the supplemental material. Those that remain are the most endpoints. I does result in improved clarity.

3.- More references needs to be used in the discussion section and more explication of the results needs to be added

Authors’ reply: Done. References to two critical meta-analyses and the studies one of them included are analysed to attempt to establish the external validity of this work. This is achieved by comparing the DES arm of this meta-analysis with the DES arms of the others, one endpoint at a time. It also compares the frequency of the three types of ACS (unstable angina, Non-ST segment myocardial infarction NSTEMI, and ST segment myocardial infarction NSTEMI, which present very different risks of adverse events during and after the coronary PCI.

4.- The conclusion section does not express any particular conclusion or any new information obtained after the discussion of this work, I suggest rewriting the conclusion and making it more useful for understanding your contribution to this area of knowledge

Authors’ reply: Done. The conclusion traces directly to the results and is an unambiguous clinical conclusion.

5.- I suggest adding more references, from the last 5 to 10 years in the discussed topics, upgrading the number to more than 60 references

Authors’ reply: Done. 60 references. However, few were published after 2014 except for meta-analyses and practice guidelines. The reason is that practice guidelines recommended DES as standard of care as of 2012 and clinicians are stopped using BMS even in trials after that. As for comparing Drug eluting stents, this has been performed but does contribute to validation process of this meta-analysis because no significant differences between DES in ACS has been demonstrated (except for bioabsorbable DES that are a completely different and recent category of implants: there is no trial comparing TiNOS with bioabsorbable DES at this time.

Round 2

Reviewer 2 Report

In my point of view, the authors have successfully attended all my comments and suggestions